# "Pretty in Pink"—The Pink Color in Architecture and the Built Environment: Symbolism, Traditions, and Contemporary Applications

**Justyna Tarajko-Kowalska [1,*]** and **Przemysław Kowalski [2]**

1   Department of Spatial Planning, Urban and Rural Design, Faculty of Architecture, Cracow University of Technology, 31-155 Kraków, Poland
2   Department of Landscape Architecture, Faculty of Architecture, Cracow University of Technology, 31-155 Kraków, Poland; pkowalski@pk.edu.pl
*   Correspondence: justarajko@tlen.pl

**Abstract:** The main goal of this article is to summarize and present the most important facts concerning the use of the pink color in the built environment of the 20th and 21st centuries, considering its symbolic, functional, and decorative aspects, with particular emphasis on Western cultures. This monograph of color is aimed to contribute to a better understanding of the place and meaning of pink in the contemporary architectural space and to allow architects to use this color with greater awareness of its characteristic features. The results of the analysis of over 100 pink buildings and spaces, collected by the authors since 2016, are grouped into seven main thematic sections, which express different ways of applying pink in the built environment: as a traditional color, a stereotypic feminine and girlish color, a contrast color in public spaces, an extravagant color, a symbol of peace, hope, tolerance, and solidarity, a trendy color, and finally an "Instagramable" and fictional color. The main conclusion is that the pink color usage in contemporary architecture is very diverse and reflects the various associations and symbolisms of the color itself, which can only be understood in its socio-cultural contexts. Currently, two opposing tendencies are especially compelling—the first related to the kitschy and plastic aesthetic of "Barbie pink", and the second associated with more neutral and universal "Millennial pink".

**Keywords:** pink; Millennial pink; Barbie pink; architecture; built environment; aesthetics; symbolism

## 1. Introduction

The pink color has a wide variety of associations, sometimes quite adverse. It is described as charming, glamorous, romantic, gentle, sweet, feminine, sensitive, and tender, but also childish, artificial, garish, and vulgar. Researchers consider the pink color divisive, polarizing, triggering of contradictory emotions, and simultaneously versatile (Blegvad 2019; Steele 2018; Nemitz 2006; Roth 2015).

The perception of this color is also not unequivocal, and depends on particular nuances, varying from delicate pastels and corals to saturated and bold "shocking" pink amaranths and magentas. Pink was isolated as a name for color only in modern times because of a particular civilizational and cultural need (Barczak 2021). The English word "pink" was used to describe light red color only in the mid-1700s (Blegvad 2019). In other European languages, the pink color name usually came from the rose flower, e.g., "rosa" in Latin, German, Spanish, and Italian, "rose" in French, "różowy" in Polish, etc. (Blegvad 2019; Barczak 2021).

The spectrum of colors referred to as "pink" is vast and covers both pure tints of red (R) as well as those with an admixture of yellow (Y80R, Y90R) or with a blend of blue (R10B, R20B, R30B), according to the Natural Color System (NCS) nomenclature (Tarajko-Kowalska 2022).

A characteristic feature of pink is that it is the only tint of a primary color with a separate name, and although it comes from red, it has utterly distinct symbolism and usage, and consequently also architectural applications, which have varied greatly across time and space (Steele 2018). The symbolism and significance of pink color is also deeply rooted in the observer's cultural background (Blegvad 2019; Nemitz 2006). For example, the pink color is important in Japanese culture, being associated with the beautiful but fleeting pink blossoms of the cherry trees, symbolizing the premature death of the samurai (Nemitz 2006). There are also many words for "pink" in Japanese language, e.g., "sakura-iro" (cherry blossom color), or more recent "pinku" used for Western pink (Steele 2018).

In the 18th century, pink color was fashionable among aristocrats and became a symbol of class and luxury in European courts. It was especially favored in France, mostly due to the mistress of King Louis XV, Madame de Pompadour (1721–1764), who was known for her predilection for pink, not only in her clothes but also in interior decorations. In 1757, a French porcelain manufacturer even developed a unique shade of pink, named in her honor "Rose Pompadour" (Steele 2018; Bucknell 2017; Barker 2015). Both the Baroque and Rococo movements created the perfect environment for pink's rise also in architecture. So, in the 18th century, pink mansions and churches were built all across Europe, and over the next century, the pink color even blossomed in popularity both in interior decoration and palaces' façades. But later, especially in Western cultures, pink was strongly feminized and eroticized for almost one century. This trend was so strong that it affected to some extent also other parts of the world, although in India and other Asian cultures pink was always a unisex color (Steele 2018). Consequently, in the 20th century, pink color became widely a symbol of womanhood, gender norms, and femininity (Blegvad 2019; Bideaux 2019). Because of those long-standing associations, today it is often identified with stereotypically understood femininity and goods intended only for girls (not only little ones). It is the color of Barbie, fairytale princesses, and unicorns. At the same time, some artistic communities want to counteract this and redefine the meaning of pink, giving it more neutral, mature, and even revolutionary connotations (Blegvad 2019; Bideaux 2019; Steele 2018; Bucknell 2017; Priya 2014). One of these "new" approaches to pink, which are no longer connected to prevalent gendered symbolism inherited from the 20th century, is formed by so-called "Millennial pink", which refers to a specific range of pale pink tones (Bideaux 2019; Mitchell 2017). The popularity of "Millennial pink" has caused pink to be increasingly perceived as a cool, powerful, and androgynous color in modern times (Steele 2018). But does this also happen in architecture? It is evident that all these cultural aspects of the pink color could not remain without influence on the rationales for its use in the built environment. Thus, the main goal of this paper is to summarize the most important facts concerning the use of the pink color in architecture, considering its symbolic, functional, and decorative aspects, with particular emphasis on Western cultures. This article is a part of a broader research project conducted by Justyna Tarajko-Kowalska since 2010 and dedicated to the use of individual hues in architecture, including their history, traditions, and contemporary application, preceded by monographs on red (Tarajko-Kowalska 2010), black (Tarajko-Kowalska 2013), white (Tarajko-Kowalska 2014), green (Tarajko-Kowalska 2015), yellow (Tarajko-Kowalska 2021), and blue colors (Tarajko-Kowalska 2023).

Despite there being many great examples of pink-colored buildings, as well as many traditions connected to this color in architecture, due to the previously mentioned stereotypic associations, pink is often disregarded for any significant and "serious" use in buildings, especially for public and representative functions. So, it was particularly essential for the authors to determine whether the popularity of "Millennial pink" has influenced in any way the approach to pink color in the architecture of the 21st century. Additionally, this paper was preceded by a study conducted by Justyna Tarajko-Kowalska in 2022 among Polish students of Architecture at the Cracow University of Technology, which showed no prejudice associated with pink color, confirming that the young architects from the "millennial" generation could possibly usher in a new era of its use in architecture (Tarajko-Kowalska 2022). Therefore, the authors hope that this color monograph will contribute to a

better understanding of the place and meaning of pink in the contemporary architectural space and allow architects to use this color with greater awareness of its characteristic features in the broader cultural context (Figure 1).

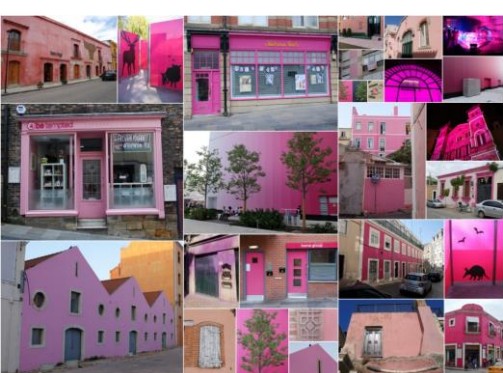

**Figure 1.** Pretty in Pink. Different applications of pink color in the built environment (collage by Justyna Tarajko-Kowalska based on her photographs, 2023).

## 2. Materials and Methods

The use of color in architecture results from a complex relationship between material, form, context, and culture seen in time. Its reasons may be utilitarian, symbolic, informative, or, finally, purely aesthetic. Therefore, all those aspects should be considered in order to understand the full diversity of functions performed by the pink color in architecture, both in the past and the present day.

The temporal scope of this article covers especially the architecture of the 20th and 21st centuries. References to older architecture only apply to those pink-colored buildings and traditions associated with pink that are related to contemporary uses of this color, where their presentation was necessary to understand current trends in this area. In the spatial context, the authors focused on the Western world in its broad sense, including Europe and both Americas. The examples from other cultural areas were used to show the globalization of the use of the pink color in architecture.

The analysis of the use of pink in architecture was conducted based on a review of relevant literature (Blegvad 2019; Bucknell 2017; Steele 2018; Roth 2015; Nemitz 2006; Barczak 2021; Priya 2014), including available references documenting the colors of traditional buildings in different regions of the world, illustrations, and descriptions found in literature and on the Internet, as well as the authors' own explorations in situ. Although the issue of color in architecture has been the subject of extensive research and a rich literature, it is difficult to find references that categorize data about single colors, especially in the case of building exteriors. Therefore, in the case of pink, monographs on this color, mainly in art, fashion, and design, were particularly helpful. The research of Kaye Blegvad (2019) was fundamental, as it presented the most comprehensive and multifaceted knowledge in the field of pink color. Also valuable sources were the books by Valerie Steele (2018) and Manuela Roth (2015), as well as articles by Alice Bucknell (2017), Elan Priya (2014), and Melissa Massello (2017).

The research on the contemporary usage of pink in the built environment was primarily based on a detailed survey of the most popular websites covering architectural news and projects from around the world, mainly, Archdaily (www.archdaily.com, accessed on 1 May 2023), Divisare (www.divisare.com, accessed on 1 May 2023; https://divisare.com/pink, accessed on 1 May 2023), Dezeen (www.dezeen.com, accessed on 1 May 2023), Designboom (www.designboom.com, accessed on 1 May 2023), and Pinterest (https://pin.it/20uEeup, accessed on 1 May 2023). Justyna Tarajko-Kowalska has already used this research method since 2010 to analyze other colors: red, green, blue, and yellow, as well as white and black. The authors have regularly reviewed the above-mentioned architectural websites since 2016, creating a collection of over 100 buildings, spaces, and small architectural objects colored

in pink. The main selection criterion was that pink, being the dominant color of the entire building/space, or its parts, determined the perception of the whole. The rich collection of pink-colored objects obtained in this way formed the basis for analyzing and determining various ways of using pink in architecture and the built environment. Of course, the authors realize that, in the case of photographs and web-based images, it is impossible to evaluate accurately the color of the buildings, but the specification and measurement of exact color notation was not the aim of the conducted research. The purpose of this study was rather to show various ways of using the broadly understood color pink in a given cultural context, considering local conditions and architects' intentions.

For a more comprehensive presentation, the authors organized the obtained results into categories/thematic sections, which, in the authors' opinion, show as many different aspects of the use of pink in architecture as possible. Based on a detailed analysis of the selected buildings, we established seven categories representing the symbolic, practical, and decorative uses of pink in both traditional and contemporary buildings, with different functions and scales.

## 3. Results and Discussion

The results of the research were grouped into seven main thematic sections, each concerning a different way of applying pink color in the architectural space. These are:

- Pink as a traditional color.
- Pink as a stereotypic feminine and girlish color.
- Pink as a contrast color in public spaces.
- Pink as an extravagant color.
- Pink as a symbol of peace, hope, tolerance, and solidarity.
- Pink as a trendy color—Millennial Pink.
- Pink as an "Instagramable" and fictional color.

### *3.1. Pink as a Traditional Color*

The first section analyzes selected traditions connected with the pink color in the built environment, which are still livable, sustained, and cultivated, also in contemporary projects. It also presents those places in the world where the application of this color in architecture was unique and therefore had an impact on the modern usages. This part collects also the iconic 20th-century buildings which are known and recognized for, among other things, their dominant pink color, thus being frequently nicknamed "pink".

#### 3.1.1. Pink Cities

This part is dedicated to so-called "pink cities", which owe their nicknames to the dominance of pinkish tones in their urban spaces. Although Jaipur is located in India, it is so widely known as "Pink City" because of the predominance of pink-colored buildings in its oldest district that it cannot be excluded from this section. Jaipur, one of India's earliest planned cities, was founded in 1727 by Maharaja Sawai Jai Singh. According to popular myth, the old town buildings started to be painted in coral-pink color in 1876, during the rule of Sawai Ram Singh I, to honor Queen Victoria's son, Albert Edward, Prince of Wales (who later became King Edward VII, Emperor of India) during his visit (Stone City Blog 2021). However, the buildings in Jaipur were painted in this color, representing welcome and hospitality in India, much earlier. The pale salmon pink paint was used on buildings to imitate the aesthetics of façades made of the much more expensive and less available pink sandstone, from which cities such as Delhi and Agra were built (Blegvad 2019). Many buildings and palaces remain painted in "Jaipur Pink", as it is known today, primarily due to urban regulations. Although the exact pink color is not specified by law, different pink nuances give Jaipur its distinctive appearance to this day. Among the most beautiful of Jaipur's pink palaces are, undoubtedly, Chandra Mahal, or City Palace, built in 1727 and the former seat of the Maharaja of Jaipur (Figure 2), as well as Hawa Mahal, also known as

the "Palace of Breeze", built in 1799 as an extension to the Royal City Palace of Jaipur for the women of the royal household (Blegvad 2019) (Figure 3).

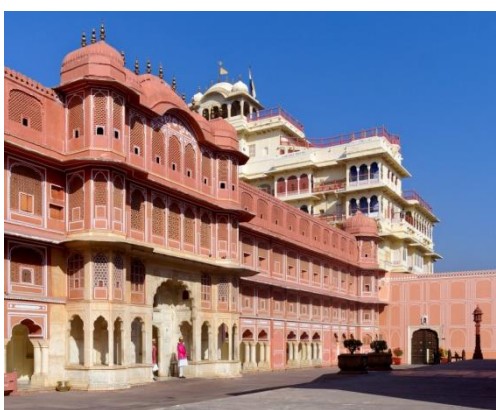

**Figure 2.** Chandra Mahal in Jaipur, India (photo by Jakub Hałun, 2019, CC BY-SA 4.0 via Wikimedia Commons).

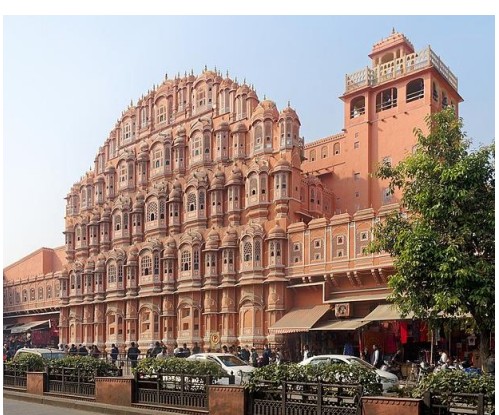

**Figure 3.** Hawa Mahal in Jaipur, India (photo by Jakub Hałun, 2019, CC BY-SA 4.0 via Wikimedia Commons).

Toulouse (France) is also known to locals as La Ville Rose (The Pink City), after the distinctive reddish-pink color of the clay bricks used to construct many of its buildings. The clay had a high iron oxide concentration, giving it a pink tint during firing. The term Ville Rose appeared officially only in 1906 on the cover of a Toulouse tourist office booklet (Stone City Blog 2021) (Figure 4).

3.1.2. Suffolk Pink

Another widely recognizable practice of using pink is connected with the unique rose colors found in abundance on the building façades in Suffolk, a ceremonial county in England. The tradition of so-called "Suffolk Pink" dates back to the 14th century, when these varying pink tones were formed by supplementing the traditional lime wash with natural substances native to the country such as ox or pig blood, sloe berries, or elderberries. Those ingredients were combined with crushed brick, burnt clay (which made red ochre), or burnt iron stones to obtain a more orange tinge. As a result, Suffolk Pink decorative colors range from a pale shell to a deep brick color. The pink-washed houses can still be found in places such as Long Melford, Lavenham, Sudbury, Hadleigh, Framlingham, Cavendish, and other county villages. Today the use of Suffolk Pink is restricted and protected by a series of bylaws and regulations. Although there is not just one specific color, councils have protected Suffolk Pink to ensure that the wrong nuances are not used (Wright 2023; Hume 2022) (Figure 5).

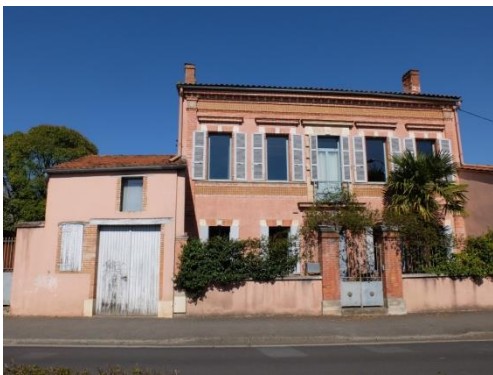

**Figure 4.** Salmon pink building in Toulouse, France (photo Justyna Tarajko-Kowalska, 2019).

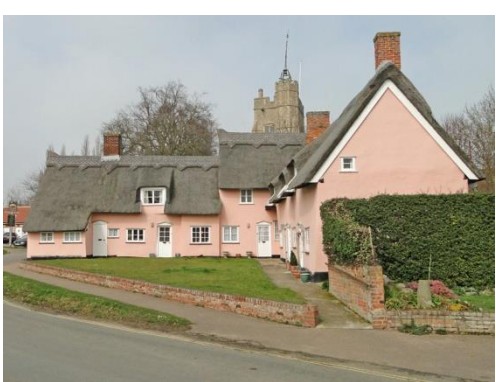

**Figure 5.** Houses in Cavendish, UK, painted in Suffolk Pink color (photo by Adrian S Pye, 2015, CC BY-SA 2.0 via Wikimedia Commons).

### 3.1.3. Mexican Pink

Another tradition of using pink color is associated with Mexico, although it can be extended to the whole of Latin America (Sanchez 2022), where this color is prevalent in craft, art, and architecture, thus forming a part of the cultural identity (Mukhopadhyay et al. 2016; Martinez 2018). The indigenous tradition of the color known as "Mexican pink" (Spanish "Rosa Mexiano") dates back to times much earlier than the name itself. The term "Mexican pink" was only coined in 1949, through the efforts of Mexican artist, cartoonist, writer, filmmaker, and fashion designer Ramón Valdiosera. Giving an interview after a fashion show for his collection inspired by traditional Mexican clothes at the prestigious Waldorf Astoria Hotel in New York, Valdiosera explained the origin of the intense shade of bougainvillea pink used in the fashion collection as an inseparable element of Mexican culture, present in toys, indigenous costumes, sweets, crafts, art, and architecture. Journalists then called this color "Mexican pink", a term which is currently used mainly for a highly saturated magenta nuance (Martinez 2018; Sanchez 2022).

The tradition of "Mexican pink" is also livable and cultivated in the architecture of that country. Especially, two 20th-century Mexican architects who widely used pink colors in their works should be mentioned here: Luis Barragán (1902–1988), winner of the Pritzker Price in 1980 and regarded as the most influential person in modern Mexican architecture, and Ricardo Legorreta (1931–2011), awarded the prestigious UIA Gold Medal in 1999, the AIA Gold Medal in 2000, and the Praemium Imperiale in 2011. Barragán, in frequent collaboration with the artist Jesús "Chucho" Reyes, often used pink in the houses he designed, especially in Mexico City. One of the nuances is even named "Barragan pink." Barragán House, the personal residence built by Barragán in 1948, where he lived and worked until his death, and which has a characteristic pink wall surrounding the roof terrace, was a manifesto of his architectural approach (Barragán House 2023) (Figure 6). House Pedregal (Casa Pedregal), designed in 1947–50, has an entirely light pink exterior

and a pink kitchen and hallways. Also, House Gilardi (Casa Gilardi), created in 1975–76, is one of the architect's undisputed masterpieces and features a bold color palette consisting of bright pink for the front and rear façades (Gilardi House 2023). However, Barragan's most famous project featuring pink is Cuadra San Cristóbal, in Los Clubes, built between 1967 and 1968 in collaboration with Andrés Casillas for Folke Egerström and his family. One of the most recognizable elements of the complex is the massive, horizontal wall finished in pink stucco, which defines the north–south axis of the plot, dividing it into two parts—one more private, and the second mainly dedicated to equestrian facilities (Cuadra San Cristóbal 2023) (Figure 7).

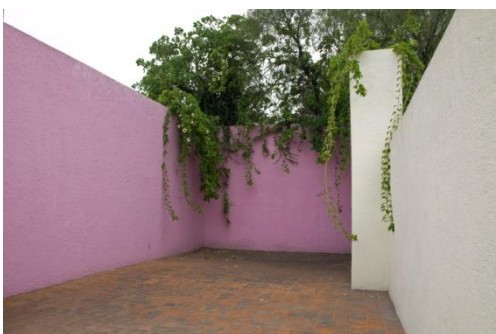

**Figure 6.** The pink wall surrounding the roof terrace of Barragán House (photo by Ymblanter, 2014, CC BY-SA 3.0 via Wikimedia Commons).

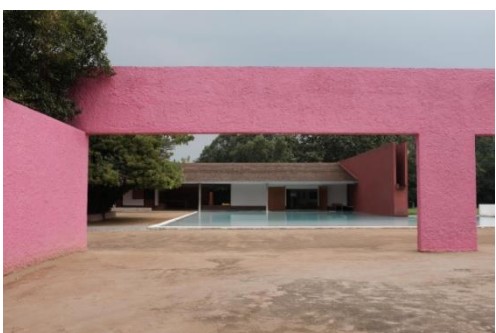

**Figure 7.** Pink stucco wall of Luis Barragan's Cuadra San Cristóbal complex (photo by Anna Bertho, 2018, CC BY-SA 4.0 via Wikimedia Commons).

In turn, the most famous project by Ricardo Legorreta with the use of pink is the Hotel Camino Real Polanco, built for the 1968 Olympics in Mexico City, whose characteristic element is a giant openwork pink wall, resembling traditional Mexican paper cuts.

Nowadays, "Mexican pink" is considered an element of national identity, and it is widely present in the clothing and textile art of native peoples, crafts, fine arts, and popular architecture (Figures 8 and 9). "Mexican pink" recently became the standard color of Mexico City's visual communication, being used, among other things, for taxis, credentials, and official documents (Martinez 2018).

Modern Mexican architects also reach for pink in their works, although it is more often the currently fashionable "Millennial pink" than the highly saturated "Mexican pink". It is especially worth mentioning three realizations of houses from the second decade of the 21st century.

The first is Monte House, designed in 2019 by TACO taller de arquitectura contextual. It is a compact vacation home designed for a couple of young adults, contrasting the wild landscape of south-eastern Mexico with the warm pink color of its geometrical shape (Monte House 2019).

Another project is Bugambilias House, designed in Merida by Taller Mexicano de Arquitectura in 2019. Monolithic pink-pigmented concrete volume rests on a grey concrete basement to generate the contrast between the two levels (Bugambilias House 2021).

The last is House in Tres Rios, designed in 2020 by Cesar Bejar Studio in Culiacan Rosales. The house is a monochrome pink structure, free of ornaments, which breaks the plasticity of the surroundings through the contrast of color, shape, and texture (House in Tres Rios 2021).

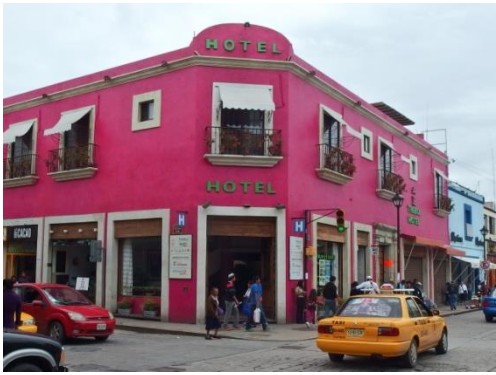

**Figure 8.** Hotel in Oaxaca colored in "Mexican pink", Oaxaca, Mexico (photo by Justyna Tarajko-Kowalska, 2014).

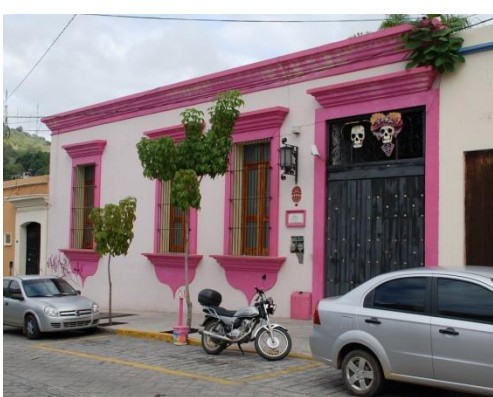

**Figure 9.** Pink details around windows and doors of a house in Oaxaca, Mexico (photo by Justyna Tarajko-Kowalska, 2014).

3.1.4. Pink Classics—Iconic Pink Buildings

This section presents the authors' selection of characteristic 20th-century buildings known for their distinctive and unique uses of pink color, due to which they are often given the nickname "pink".

Casa de Serralves, known locally as "Casa Cor de Rosa" or "Pink House", is a villa and museum located inside the Park of Serralves in Porto, Portugal. It was commissioned by the second Count of Vizela, Carlos Alberto Cabral, and built by the architect José Marques da Silva between 1925 and 1944 (Blegvad 2019). The pink color, chosen for façades as well as for the garden surfaces and the edging walls of the water channels, was dreamed up by the French architect Charles Siclis, whom Cabral also consulted. He sent the Count two watercolors depicting the façades of the villa with all the elements of the design tinted pink. That became the starting point for the final color palette of this outstanding example of French Art Deco architecture (Leevers 2020). Today, Casa de Serralves serves as the headquarters of the Serralves Foundation (Fundação de Serralves) and is an important extension of the Museum of Contemporary Art, used mainly for temporary exhibitions (Figure 10).

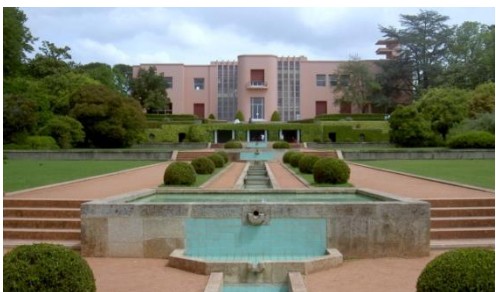

**Figure 10.** Casa de Serralves in Porto, Portugal (photo by randwill, 2003, CC BY 2.0 via Wikimedia Commons).

During the interwar period, pink color became very popular in the architecture of the United States. At that time, many buildings in Art Deco style were finished in coral or pink pastel stucco, partially inspired by the peachy-pink colors typical of the traditional architecture of Havana and other Cuban places (Massello 2017). Among the famous pink-colored buildings from the 1920s is the hotel The Don CeSar Beach Resort and Spa, also known as the "Pink Lady", designed by Henry H. Dupont in 1928 and located in St. Pete Beach, FL, USA (Massello 2017) (Figure 11).

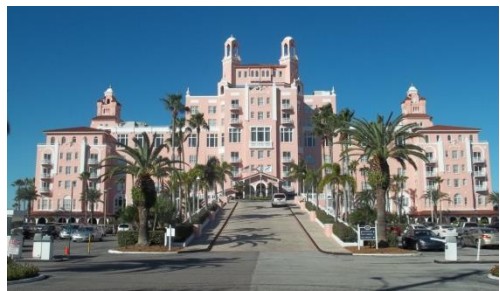

**Figure 11.** "Pink Lady"—The Don CeSar Beach Resort and Spa in FL, USA (photo by Ebyabe, 2011 CC BY-SA 3.0 via Wikimedia Commons).

Another outstanding pink hotel is undoubtedly The Royal Hawaiian Hotel, on Waikiki Beach in Honolulu, on the island of Oahu, Hawaii. It was designed in a Spanish Revival and Moorish-inspired style by the renowned architectural firm Warren and Wetmore in 1927. The nickname "Pink Palace of the Pacific" was given thanks to the hotel's prominent location and the pale pink color of the stucco façades. The color was chosen presumably by a landscape architect to match the exotic setting and contrast with the sea's blue and the surrounding greenery (Massello 2017) (Figure 12).

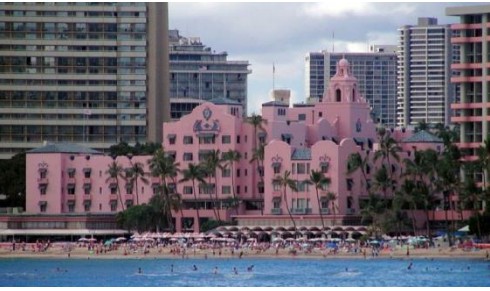

**Figure 12.** "The Pink Palace of the Pacific"—The Royal Hawaiian Hotel in Hawaii (photo by Gerald Farinas, 2004 Public domain, via Wikimedia Commons).

A list of iconic buildings colored in pink must include The Pink Palace Mansion (Los Angeles, CA, USA), purchased in 1957 by the famous actress Jane Mansfield and her husband, Mickey Hargitay. The Mediterranean-style mansion was fully painted pink,

which became synonymous with the actress. It was, according to her, a "pink landmark". The Pink Palace was sold after her premature death in a car accident in 1967 (Blegvad 2019). Although the house was demolished in 2002, its aesthetics are still an inspiration for many contemporary buildings, for whose owners, as for Mansfield, pink is more than just a color, becoming an expression of their lifestyle and a feature distinguishing them from the environment and society (e.g., the houses of Jamie Nelson and the House of Adora presented in Section 3.3).

Although pink might seem an atypical choice of color for high-rise buildings, architects from the renowned office of Skidmore, Owings, and Merrill (SOM) met this challenge by erecting two such skyscrapers in the 1980s. The Georgia-Pacific Tower, built in 1981 in downtown Atlanta, GA, USA, is a more than 200 m high modernist skyscraper, clad in pink granite quarried from Marble Falls, Texas (Georgia-Pacific Tower 2015) (Figure 13). The US Bancorp Tower (also known as the Big Pink) is a 42-story skyscraper more than 160 m high in Portland, OR, USA. The building, designed by SOM together with Pietro Belluschi in 1983 as the headquarters for US Bank, is distinguished both by its shape and color. The exterior is covered by carefully selected pink granite quarried in Spain. The Pittsburgh Plate Glass used for the windows was finished in a semitransparent coating of copper and silver that looks pink from the outside (Libby 2016) (Figure 14).

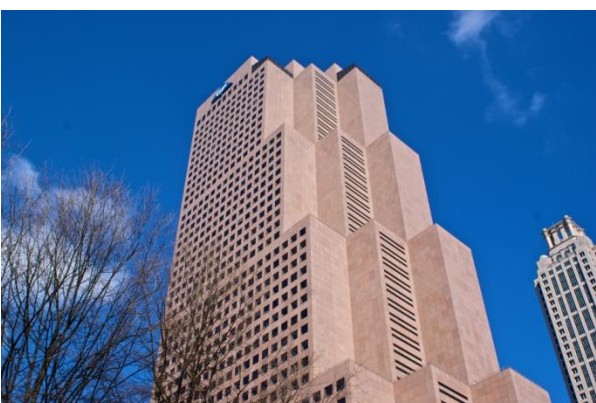

**Figure 13.** The Georgia-Pacific Tower, Atlanta, GA, USA (photo by Connor.carey, 2010, CC BY-SA 3.0 via Wikimedia Commons).

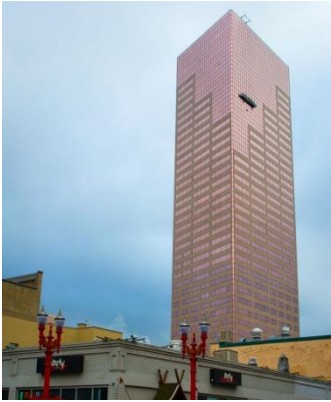

**Figure 14.** "Big Pink"—The US Bancorp Tower, Portland, OR, USA (photo by Visitor7, 2013 CC BY-SA 3.0 via Wikimedia Commons).

Although the famous architect Frank Lloyd Wright (1867–1959) did not use pink very often, his use of that color was as ahead of his time as his architectural works. According to some sources (Massello 2017; Cills 2017), pink was among the colors proposed by Wright for The Guggenheim Museum (New York, NY, USA). Although that building, finished shortly after the architect's death in 1959, is white-colored today, Wright's watercolor

paintings show different color variations, including the pink one. Two pink colors, Wright Shell Pink and Wright Flesh Pink, were also included in the "Taliesin West color palette", a set of 36 colors inspired by the natural landscapes of Arizona, developed by Wright for Martin Senour, a West Coast paint brand, in 1955 (Massello 2017; Cills 2017). Wright actually completed two pink-colored buildings. Pinkish, earthy tones distinguish the Grady Gammage Memorial Auditorium in Tempe, AZ, USA (1964) (Figure 15), and pale pink is the distinctive color of the stucco walls of the Marin County Civic Center ("Big Pink") in San Rafael, CA, USA (1957) (Frank Lloyd Wright Foundation 2023). It is also worth mentioning the light pink volumes of The King Kamehameha Golf Course Clubhouse in Waikapu, Maui, HI, USA (1993), designed by John Rattenbury but based on Wright's 1957 project of an unbuilt house for Marilyn Monroe and her husband Arthur Miller (Massello 2017).

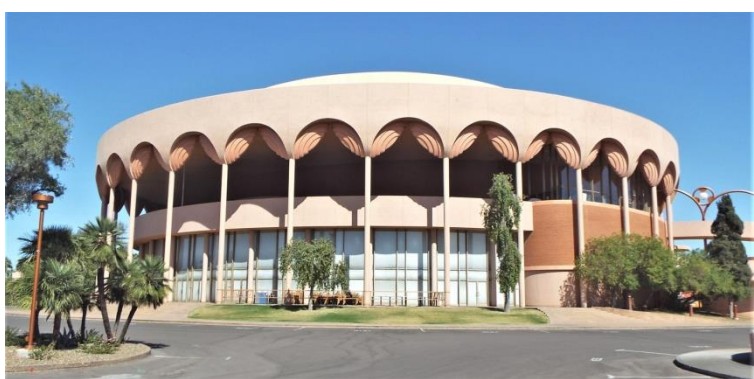

**Figure 15.** Grady Gammage Memorial Auditorium in Tempe, Arizona, USA (photo by Marine 69–71, 2016, CC BY-SA 4.0 via Wikimedia Commons).

But an exterior pink color was chosen not only for the façade finishes of luxury hotels and extravagant villas. One of the examples of the unique uses of this color in industrial architecture is the circulation tank Umlauftank 2—UT2, also known as Rosa Röhre (Pink Tube), located in Berlin, Germany. The flow circulation channel of the Laboratory for Hydraulic Engineering and Shipbuilding (Versuchsanstalt für Wasserbau und Schiffbau) at the Technical University of Berlin was constructed between 1968 and 1974 and designed by the architect Ludwig Leo and the engineer Christian de Boes. The box-shaped laboratory was clad in aluminum panels in blue, while the giant tube was coated with a thick layer of PU foam and painted pastel pink. Stylistically, the building, listed since 1995, belongs to post-war modernism and the international avant-garde, being called an icon of pop art architecture. From 2014 to 2017, the architect HG Merz renovated it on behalf of the Wüstenrot Foundation, and the already faded colors were carefully restored (Umlauftank 2023; Rosa Röhre 2023) (Figure 16).

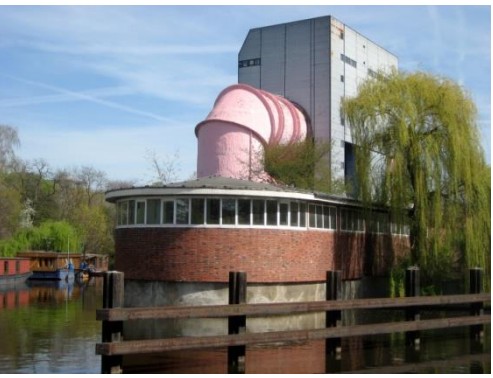

**Figure 16.** Umlauftank (Rose Rohre) in Berlin, Germany (photo by Ice Boy Tell, 2007, CC BY-SA 4.0 via Wikimedia Commons).

### 3.2. Pink as a Stereotypic Feminine and Girlish Color

This part presents projects where pink emphasizes typical, stereotypical associations of that color with femininity and eroticism, but also little girls and Barbie dolls.

While pink has almost always been associated with delicacy, flowers, and the sweet scent of roses, its strong association with femininity in Western cultures is relatively recent. Gender-coded references to colors appeared at the end of the 19th century and intensified after World War II. Although nowadays the assignment of pink for girls and blue for boys seems obvious, in 1918 it was the opposite. Blue, as more delicate and dainty, seemed more suitable for girls, and pink, being simply a light version of red, was more appropriate for boys (Del Giudice 2012). The change took place only after World War II. History connects it especially with the United States, where thanks to the influence of "Mamie" Eisenhower (Mary Geneva Doud Eisenhower), the wife of President Dwight Eisenhower and the First Lady of America in the years 1953–1961, the color pink became a symbol of a woman as a housewife, devoted to her family (Mitchell 2017). "Mamie" Eisenhower was pivotal in popularizing the color, which is often referred to as "Mamie Pink" or "First Lady Pink". She also re-decorated the private quarters in the White House in pink so that reporters called it the "Pink Palace". Her influence on interior decoration trends was so significant at that time that, according to sources, one in four homes in the US had pink bathrooms; pink kitchens were also popular (Stone City Blog 2021). During that time, increased sales of gender-dedicated products promoted the specific coding of pink-for-girls and blue-for-boys colors. Finally, pink was assigned to girls and all items related to them—clothes, toys, school supplies, etc. Nowadays, the vast supply of pink products dedicated to girls is so ample that it makes it possible to build an almost entirely pink world for them, in which all equipment is in this color. This phenomenon was noted by, among others, the artist JeongMee Yoon in The Pink & Blue Project (Yoon 2005), for which she was photographing Korean and American girls and boys in their bedrooms, surrounded by the abundance of their pink and blue belongings (Yoon 2005).

Moreover, modern research confirms that pink color is actually preferred by girls starting from age two and continuing throughout preschool, while boys of that age show increasing avoidance of that color (LoBue and DeLoache 2011). Regardless of the root cause, and whether girls naturally favor the color pink or it is culturally conditioned, in the 20th century it became widely associated with little girls and all things feminine (Steele 2018). Additionally, pink toys, especially dolls, with the famous Barbie at the forefront, but also other pink characters such as the friendly cat Hello Kitty or even Peppa Pig, together with the pink merchandise connected to them like dolls' houses, etc., made pink a color that is, on the one hand, associated with fairytales and playful, but on the other hand, infantile, immature, and artificial, or associated with plastic.

Many of these associations have found reflections in architecture and the built environment. For example, one of the projects in which the pink color is very gender-associated is "His house and Her house", a renovation of two houses in Dameisha Village, carried out by the Chinese architecture studio Wutopia Lab for the 2017 Bi-City Biennale of Urbanism/Architecture in Shenzen, China. Architects painted "Her house" pink, filling it with flowers, and "His house" blue, filling it with meat decorations, to discuss the contrasting roles of men and women in the kitchen and how these are reflected in their dietary habits (Block 2018).

Some of the projects, which may border on kitsch, refer to the dream of some adults, mainly women, of spending time in a pink house. One such project, reminiscent of a full-scale dollhouse, is Eaton House Studio (The Pink House) in Tiptree, UK, painted pink in 2012 to attract guests with a controversial look. The façades and the interiors of this house for events are finished in pink, thus allowing visitors to feel like a fairytale princess (Eaton House Studio 2023).

One project that recalls the association between the color pink and girlhood/childhood in an unusual way is the Hu Huishan Memorial, designed by Jiakun Architects in 2009 in Anren Town, China. The memorial hall was built in memory of Hu Huishan and other students killed by the Wenchuan earthquake in 2008. The monochromatic grey volume of the building is modeled on the form of the tents that are commonly erected after an earthquake. In contrast, the entire interior is painted pink, Hu Huishan's favorite color, symbolizing a lost childhood and an innocent life that ended prematurely (Hu Huishan Memorial 2013).

Nevertheless, some buildings fit the stereotypical meaning of pink color by using a cartoon-like, kitschy aesthetic. This is the case with the facilities dedicated to famous characters such as Hello Kitty or Barbie. However, the specific artificial style of these buildings is accepted and even favored by fans due to the possibility of finding themselves in a real space modeled on the world of a favorite character.

Hello Kitty is a popular cat-like character, invented in 1974 by Yūko Shimizu of the Japanese company Sanrio, which is the heroine of animated films and computer games, but also decorates many gadgets, mainly in pink color. Its long-term popularity has also resulted in the emergence of Hello Kitty Cafes, where the pink color prevails in both the buildings' exteriors and interiors.

In 2018, the first Hello Kitty Grand Café was opened in Irvine, CA, US, and the following year another opened also in Las Vegas, NV, USA (Hello Kitty Grand Café 2023). Kitty enjoys exceptional popularity in Japan and Korea, where in many cities there are stationary cafes dedicated to the fans of a friendly cat, such as Café de Miki with Hello Kitty in Himeji, Japan (Figure 17).

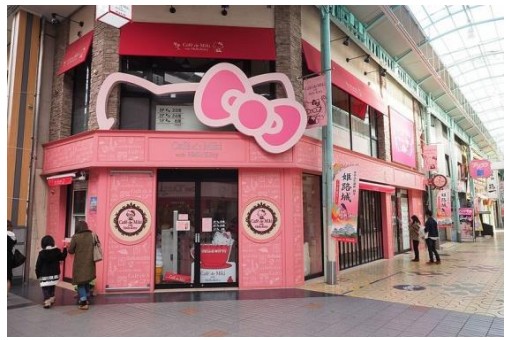

**Figure 17.** Café de Miki with Hello Kitty in Himeji, Japan (photo by othree, 2018, CC BY 2.0 via Wikimedia Commons).

The famous Barbie doll has been one of the world's most popular toys since it was launched in 1959 by Mattel, the US toy manufacturing company. Originally, the Barbie doll was designed as an elegant and beautiful fashion icon who always dressed according to new style trends. But in the 1970s, Barbie's aesthetic mainly became pink (Steele 2018). Nowadays, this color is referred to as "Barbie Pink" and is standard not only for the clothes of the popular doll but also in the interior design of her houses and their equipment. According to Valerie Steele, "Barbie pink" is not only a feminine and childish color but also has a kind of fake, artificial quality, which can be problematic or fabulous at the same time (Steele 2018).

Obviously, the color pink also dominates the architecture associated with Barbie.

The first-ever Barbie Flagship Store for Mattel was designed in 2009 by New York-based studio Slade Architecture in Shanghai, China. The building hosts the world's largest and most comprehensive collection of Barbie dolls and licensed products, as well as a range of services and activities for all Barbie fans. Although the exterior is not pink-colored itself, the pink interior is visible through the transparent panels, thus giving the impression of a pink facade, especially at night (Barbie Shanghai Store 2009).

Taiwan's first Barbie Café, a pink restaurant based on the Barbie theme, authorized and owned by Mattel, was opened in Taipei in 2013. From the outside, pink signboards invite passers-by to the restaurant, and the entrance was made in the form of a pink glass cuboid. The interiors are also dominated by pink—present on walls, furniture, and decorations (De Lacey 2013).

The building that caused the most controversy and became the cause of feminist demonstrations was The Barbie Dreamhouse Experience, an interactive exhibit accessible for four months in 2013. It was designed as a joint venture between Austria's EMS Entertainment and Mattel in Berlin, Germany. The building was modeled on the Barbie-doll Dreamhouse, and one hundred gallons of pink paint were used to paint it. Feminist groups opposed the attraction, saying the Barbie doll and pink color stereotype women, reducing them solely to external appearance. Despite the protests, the installation enjoyed great interest and was visited by about 3000 people daily (Barbie The Dreamhouse Experience 2013; Engel 2013) (Figure 18).

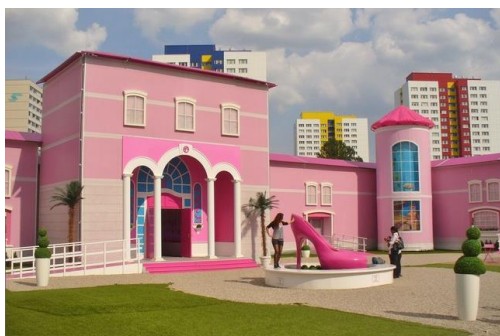

**Figure 18.** The Barbie Dreamhouse Experience, Berlin, Germany (photo by Colin Smith, 2013, CC BY-SA 2.0 via Wikimedia Commons).

Interestingly, Mattel and Airbnb created a real Barbie Dreamhouse in Malibu in 2019 to celebrate Barbie's 60th anniversary. Initially a dollhouse, the Barbie Dreamhouse debuted in 1962 as a surprisingly modest studio apartment with modern furniture. Over the years, it followed the trend of suburban development, today being a pink three-story house—a kitschy, plastic miniature of the American "dream house". The real Dreamhouse in Malibu could only be rented for two days in October 2019. Airbnb donated the rental income to Barbie's Dream Gap initiative, organized on behalf of the GoFundMe fundraising platform, which aims to help young girls regain faith in their abilities and not give up on their dreams (Mankiewicz 2019).

But pink color in architecture is associated with femininity not only in the context of little girls and Barbie but also eroticism. According to Barker, pink is first and foremost the color of love and sexuality, and hence it implies a view of women that is centered on these qualities (Barker 2015).

One example is the Pink Palace (former Eros-Center), a brightly pink-colored sex house in the St. Pauli district of Hamburg, Germany. Eros-Center, built in 1967 by the businessman Willi Bartels (1914–2007), was meant to rid the streets of prostitutes soliciting passers-by. In the 1960s, it was Europe's largest and most modern brothel. Today, the renovated building has a new pink facade color and name, the Pink Palace, but its function has remained unchanged (König von St. Pauli 2007) (Figure 19).

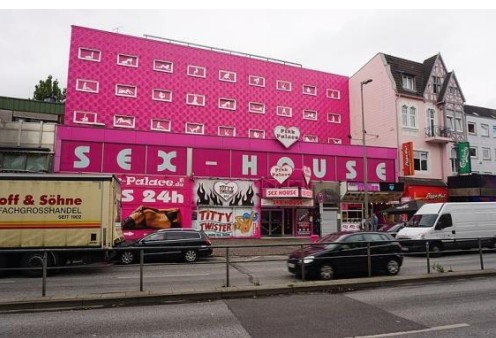

**Figure 19.** Pink Palace (former Eros-Center) in Hamburg, Germany (photo by Andrew Milligan sumo, 2019, CC BY 2.0 via Wikimedia Commons).

### 3.3. Pink as a Contrast Color in Public Spaces

This section deals with the pink color that architects and artists use to visually contrast designed objects with their surroundings. That is especially true when the goal is to create a visual landmark or cause a temporary spatial installation to stand out in space. Because the pink color in nature occurs mainly in flowers, monochrome pink structures are particularly contrasting in those locations where greenery predominates, whether in urban or open landscapes.

An example of such use of pink is the project of a summer house, Vila Hermína, designed in 2009 by HSH architekti in Jevišovice, Czech Republic. The small cubic volume of the building is fully covered with a polyurethane-spray pink-colored coat, which also performs thermal and waterproofing functions. The finishing material and color were selected as a tribute to the architects' favorite Umlauftank (Rose Rohre) building by Ludwig Leo (see Section 3.1.3). As a result, the building looks like a giant pink flower set in the mostly greenish and brownish colors of the surrounding rural landscape (Vila Hermína 2009).

Pink-colored buildings and objects also stand out strongly against the backgrounds of grey and dark, subdued colors that dominate the architecture of many cities. For example, in the 2017 Impostor project, Les Malcommodes architects used pink color to create a distinctive sculptural element in contrast to a highly touristic environment in Quebec, Canada. The pink monolithic tunnel generated the illusion of a crossing through a fortification which was initially built in the 17th century at the St. Lawrence River and reconstructed in 1977 (Impostor 2017).

Among small architectural objects, pink sculptures especially evoke a unique effect in public spaces. Two such projects in particular are worth mentioning.

The first is the giant sculpture of Pink Rabbit located in Vienna, Austria, created by Ottmar Hörl (born 1950), a German conceptual artist. It is a 3D interpretation of Albrecht Dürer's famous watercolor painting "Young Hare", widely acknowledged as a masterpiece of observational art and housed in the Albertina Museum in Vienna. The Pink Rabbit was designed as a tribute to Dürer on the 500th anniversary of his famous painting from 1502. In 2014, the bright pink polyester sculpture was placed on the roof of the Albertina Museum during an exhibition of the "Young Hare" (which is only shown once a decade) and then moved to its current location in front of the Vienna State Opera. According to Albertina director Klaus Albrecht Schröder, the principal intended signaling effect of Hörl's sculpture was that "Art is artificial", which was achieved by the enormous scale and unnatural pink color of the rabbit (Dürer-Hare for the Albertina 2014).

The second example is the three light pink silos marked "Cocoa", "Milk", and "Sugar" located on the grounds of Malley's Chocolate Factory in Cleveland, Ohio, USA. Erected at Brook Park in 2011, the silos are today purely decorative, although they were initially planned to store milk, cocoa, and sugar, respectively. The distinctiveness and monumentality of the pink-colored silos, each measuring 3.5 m wide and 27 m tall, have turned them into a local landmark, today often used for wayfinding in the area (Cocoa and Silos 2023).

Another way to attract observer attention and change the usual perception of the environment is the introduction of a pink surface to a public space. Two such projects especially deserve to be mentioned here. The first is Pink Street (Rua Cor-de-Rosa) in Lisbon, Portugal, designed by Jose Adrião Arquitectos in 2011. Pink Street, originally Nova do Carvalho Street, is located in the Cais do Sodre district, close to Lisbon's main square, the Praca do Comercio. For many years, the area was a place of disrepute with a concentration of brothels, cheap bars, and nightclubs. In 2011, with the participation of the city authorities and the Associação do Cais do Sodré, a project was carried out to renovate the street and the buildings along it. As a result, the brothels and strip clubs were liquidated, the road was closed to car traffic, and finally, to emphasize the change, its surface was painted pink. Thanks to this, Pink Street has gained an entirely new dimension and is currently a dynamic public space that is inclusive, open, and multifunctional. It is a meeting place full of bars, restaurants, and pubs, and due to its striking color it is also one of Lisbon's most popular Instagram spots (Pink Street 2013) (Figure 20).

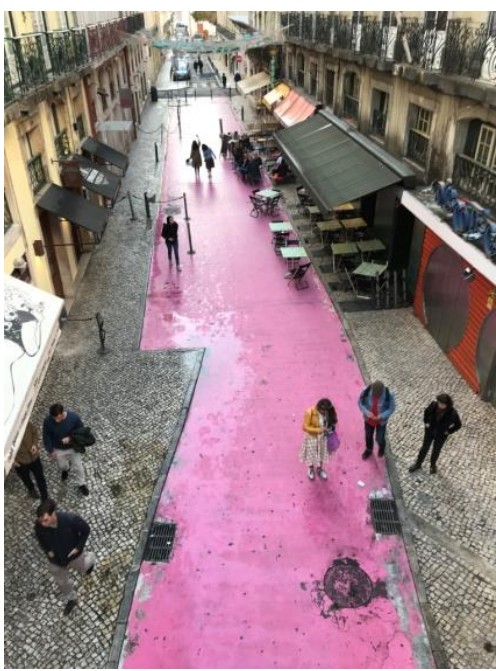

**Figure 20.** Pink Street in Lisbon, Portugal (photo by Carlos Felipe Pardo, 2019, CC BY 2.0 via Wikimedia Commons).

The next pink surface intervention in public space was LightPathAKL, by Monk Mackenzie Architects and Landlab, built in Auckland, New Zealand, in 2015. LightPathAKL is a 600 m long cycling path that complements Auckland's inner city cycling network. The former asphalt surface of a disused road was replaced with highly vivid and provocative pink resin and aggregate surfacing. This transformed the road into a contemporary urban space willingly used both by cyclists and pedestrians (LightPathAKL 2016).

Because of its contrasting properties, the pink color is also eagerly used by artists and designers in temporary installations meant as visual attractors in urban spaces and as signals of important causes. Five different examples illustrate such uses of pink.

The Pink Ghost installation temporarily transformed Cour des Ecuries into an outdoor lounge room at Rue de Furstemberg in Paris, France. The Péripheriques Archtiectes agency created this project within the Exposition of Contemporary Art in Saint-Germain-des-Près/Parcours/Paris-Soho, which took place in 2002 and was organized by the Saint-Germain-des-Près committee. Architects transformed part of the square by covering, with pink-colored resin, its surface together with four trees and streetlamps up to 2.5 m. Pink

chairs and coffee tables completed the urban furniture. As a result, the square became a specific exterior lounge meant to question the city's public space status (Pink Ghost 2015).

"Beware of Color" was a guerrilla art project using swooshes of pink paint on dilapidated heritage buildings to highlight the housing crisis in Johannesburg, South Africa. In 2014, the New York-based artist Yazmany Arboleda, together with 30 local artists, selected some precious buildings such as Shakespeare House, Clegg House, and New Kempsey, and they splashed them with pink color to draw attention to the neglect of much of Johannesburg's prime real estate. Although the project was strongly criticized by local authorities, architects, and monument conservators, the artist believed that the purpose of the performance had been achieved, as it started a discussion about buildings that have been abandoned and unnoticed for many years (Liston 2014).

Look! Look! Look! was a pink sculptural installation located in the walled garden of Berrington Hall, a historic Georgian mansion in Herefordshire, UK. The pavilion, created in 2017 by internationally renowned artists Heather and Ivan Morison, was inspired by the popularity of garden buildings in the Georgian era. The pavilion was built as a timber structure overlaid with a special woven fabric. Its delicate pink color was chosen from a traditionally Georgian palette (Look! Look! Look! 2017).

Pink Pond was a picturesque architectural installation, and the winner of the National Gallery Victoria's 2021 Architecture Commission in the Grollo Equiset Garden in Melbourne, Australia. The installation, entitled pond[er], was designed by the architecture firm Taylor Knights in collaboration with the artist James Carey. The shallow pond's water was colored pink, referencing the inland salt lakes in Victoria and highlighting the importance of water as a natural resource (Pink Pond 2021).

The Podium was a 29 m high platform built on the roof of the Het Nieuwe Instituut in Rotterdam, The Netherlands, accessible via a 143-step external staircase. Designed by renowned architects MVRDV, The Podium formed a temporary meeting place during the summer of 2022, dedicated, among other things, to various activities connected with the Rotterdam Architecture Month Festival. Both surfaces of the 600 m² platform and the giant stairs were finished in striking pink to increase their visibility (The Podium 2022).

*3.4. Pink as an Extravagant Color*

This section discusses the extravagant and unusual use of pink color in architecture. Kaye Blegvad wrote in "The Pink Book" that pink is a very "out-of-the-ordinary" color for architecture, and because of that, pink buildings convey a sense of wonder and the extraordinary (Blegvad 2019). The buildings presented here undoubtedly meet this criterion of uniqueness.

Casa de los Milagros (House of Miracles), designed by Mexican architect Danilo Veras Godoy in Xalapa, Veracruz, Mexico, is unusual both because of its color and form. The house, built in stages in the years 1995–2002, was designed to meet the needs of a single mother and her two young children. Its organic volume with unexpectedly shaped windows, resembling a giant shellfish or crab, is covered by light-pink-colored mosaic glass (Arellano 2022).

Another extraordinary pink-colored building is the Grüne Zitadelle von Magdeburg (Magdeburg's Green Citadel), a postmodern residential complex in Magdeburg, Germany, finished in 2005, five years after the death of its famous architect, Friedensreich Hundertwasser (1928–2000). Hundertwasser's innovative architecture replaced one of the first prefabricated concrete slab buildings in the GDR during the transformation of Magdeburg's Cathedral Square. The pink-colored multifunctional building (although called "green") is being described as an "oasis for humanity and nature in a sea of rational houses" (Grüne Zitadelle 2023) (Figure 21).

The Pink Zebra Restaurant, aka Feast India Co. (FIC), is an extravagant pink-colored building inspired by Wes Anderson's aesthetic and designed in 2018 by Renesa Architecture Design Interiors Studio in Kanpur, India. The main idea of the architects was to create a distinct architectural style that stood out by using a striking color palette. The ubiquitous pink-

ness of the façades and interiors has been intersected with multi-directional stripes of white and black, resulting in surrealist and bizarre architecture (The Pink Zebra Restaurant 2020).

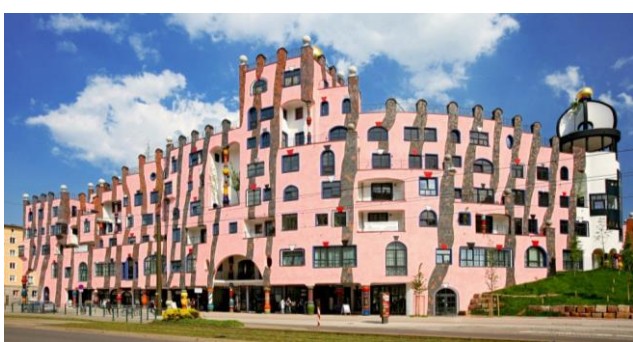

**Figure 21.** Grüne Zitadelle (Green Citadel) in Magdeburg, Gemany (photo by Mbn 1969, 2016, CC BY-SA 4.0 via Wikimedia Commons).

Pink is also often chosen for buildings whose extravagant owners want to stand out from their neighborhood. It is worth mentioning the Palazzo Chupi, a residential condominium designed in 2007 by artist, painter, filmmaker, and designer Julian Schabel in New York, USA. The building, named after the artist's wife's dog, was also inspired by the popular Spanish lollipop "Chupa Chups". The eccentric building, also the residence and atelier of Schnabel himself, was designed in the style of a Venetian palazzo and built on top of a former horse stable from 1915. It is characterized mainly by its bold pink color and has been regarded as an instant icon, an "extension of Schnabel's ego", and an "exploded Malibu Barbie house" (Lodi 2020) (Figure 22).

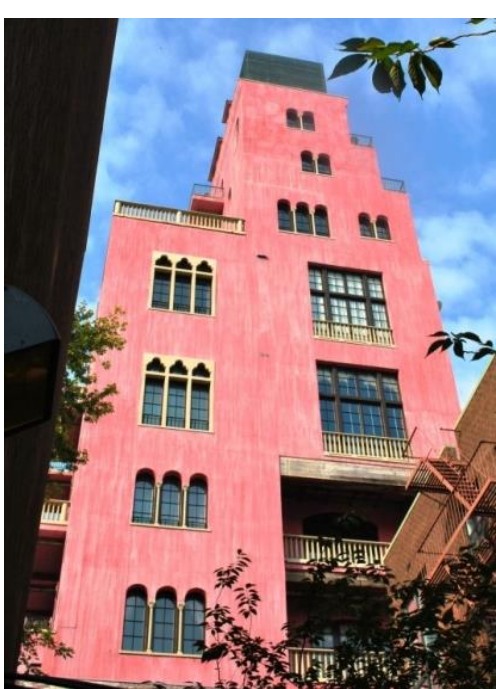

**Figure 22.** Palazzo Chupi in New York, USA (photo by Elisa.rolle, 2013, CC BY-SA 4.0 via Wikimedia Commons).

The following three projects are single-family houses, completely repainted pink by their eccentric owners due to their particular preference for this color.

The first one is the house of Emilio Rodriguez in Pflugerville, TX, USA, which in 2018 was painted entirely pink, including the roof. Pink has become Rodriguez's favorite calming color since he was paralyzed at the age of 4 in a car accident. That is why he decided to



paint pink, not only the house, but also the driveway and the fence of his property. Despite open opposition from neighbors comparing the house color to the popular pink digestive medicine Pepto-Bismol, the eccentric owner is happy with the changes and finds himself comfortable in the pink-colored environment (Careaga 2019).

The next one is the house of a fashion photographer, Jamie Nelson, who in 2019 transformed her Los Angeles (CA, USA) mansion from 1968 into a realistic version of Barbie's Dreamhouse. The pink aesthetics of the house were also inspired by the iconic Pink Palace of Jane Mansfield (see Section 3.1.4) (Fraser 2019). Also, the 2017 renovation of the Adora house in Nashville (TN, USA) was motivated by the Barbie aesthetic, so the exterior was painted totally pink (Breaux 2022).

### 3.5. Pink as a Symbol of Peace, Hope, Tolerance, and Solidarity

Pink color is widely used as a symbol of love, tolerance, peace, hope, and solidarity.

It is worth mentioning in this context the famous Casa Rosada (Pink House), known officially as the Casa de Gobierno (Government House) or Palacio Presidencial (Presidency Palace), the official residence and workplace of the President of Argentina in Buenos Aires. The multi-stage construction lasted until 1898, under the direction of many architects: the Englishman Edward Taylor, the Swedes Carl Kihlberg and Henrik Aberg, and the Italian Francesco Tamburini (Casa Rosada 2019). The exterior of the building was painted pink during the presidency of Domingo Faustino Sarmiento (1868–1874). One of the stories about that decision was that Sarmiento was trying to diffuse political tensions by mixing the colors of opposing political parties: the red of the Federalists and the white of the Unitarians. An alternative explanation suggests that the initial paint contained cow's blood to prevent damage from the effects of humidity. Nevertheless, the original pink color of the façades has remained up until today, having become the hallmark of the building (Blegvad 2019) (Figure 23).

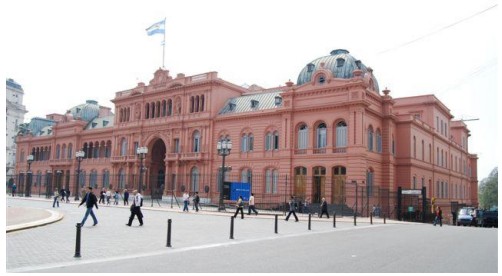

**Figure 23.** Casa Rosada in Buenos Aires, Argentina (photo by Juan Geracaris, 2007, CC BY 2.0 via Wikimedia Commons).

Nowadays, pink is a symbolic color used, among other things, by groups involved in issues important to women, e.g., activism for women's rights (like the pink sari revolution or the pussy hats), and awareness for specific causes connected with femininity. The pink ribbon, the international symbol of breast cancer awareness since 1991, is the most recognizable example of that. The pink color was chosen partially because of its strong association with womanhood. Breast Cancer Awareness Month (BCAM) has been celebrated every October for the last 90 years to raise awareness about breast cancer, which is the most commonly diagnosed cancer globally. It is also called "Pink October" because, on those days, people around the world wear pink-colored clothes or display a pink ribbon in solidarity with women suffering from that disease (Breast Cancer Awareness Month 2023). During "Pink October", buildings all around the world are also lit up in pink color. Among them are the Eiffel Tower (Paris, France), lit up pink since 2014 (The Eiffel Tower 2022) and the White House (Washington, DC, USA) (Figure 24).

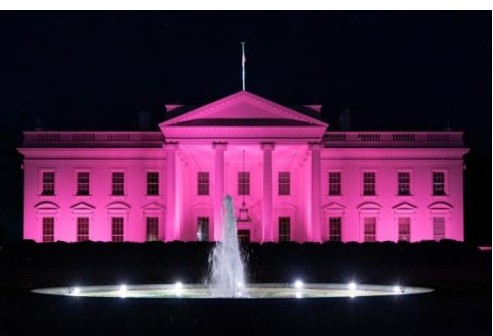

**Figure 24.** The White House in Washington is lit pink for Breast Cancer Awareness Month in October (photo by Office of the First Lady of the United States, 2021, Public domain, via Wikimedia Commons).

There are also many other initiatives around the world related to the BCAM commemorations. For example, manufacturers offer limited series of pink-colored products at this time and sell them with the promise to donate a portion of the total profits to the selected country's cancer research societies and foundations. As part of such a campaign, Icelandic farmers can choose pink plastic to wrap their hay bales instead of the traditional white or light green. As a result, the popularity of pink hay bales has increased significantly in recent years, thus creating a contrasting color accent in the harsh Icelandic countryside and, at the same time, being a vivid reminder of the cause they are supporting. Based on the popularity of the pink hay bales, the Reykjavik-based designer Tobia Zambotti proposed in 2019 an art installation called "Contemplerary". The inspiration was taken from classical Greek temple architectural ruins, but the "temple" was made from 294 pink hay bales: 90 circular for the columns and 204 rectangular for the base. The design, aimed to contrast radically with the landscape, was meant to go viral on social media in order to spread awareness and support the fight against breast cancer (Myers 2019).

Pink is also a color connected with the LGBTQ+ community, and the pink triangle has become a symbol of AIDS awareness and the modern gay rights movement. The origin of this dates back to the days of Nazi Germany, where prisoners in concentration camps accused of homosexuality were forced to wear a pink triangle. Today, this color is used for many initiatives and activities related to homosexual communities, symbolizing awareness and tolerance.

For example, the award-winning Pink Balls installation by landscape architect Claude Cormier in Montreal, Canada, was designed for the 5th Annual Creative Spaces Event in 2012, organized together with a celebration of the 30th anniversary of the creation of Montreal's Gay Village. Cormier, also known as the author of The Lipstick Forest, a pink winter garden at the Palais des congrès de Montréal (2002), proposed a delicate vault made of pink balls, hanging over San Catherine Street for a length of one kilometer (Furuto 2012). The 170,000 plastic balls, in three different sizes and five subtle pink colors, were strung together with bracing wire and crisscrossed the street, thus creating a delightful visual effect during summer events (Pink Balls 2012).

In the next presented object, pink color is a symbol of tolerance. The Bridge of Tolerance on the Odra River in Głogów, Poland, is a steel construction from 1917, rebuilt in the late 1940s after its destruction during World War II. In 1996–97, it was renovated according to the design of B. Hejduk. During the renovation, the bridge was painted in a heather-pink color chosen by TV viewers in a contest which architect Andrzej Leszek Szczypień initiated. The bridge, popularly known as "Pink", received the official name "Bridge of Tolerance" in 2005 (Maciuszczak 2018) (Figure 25).

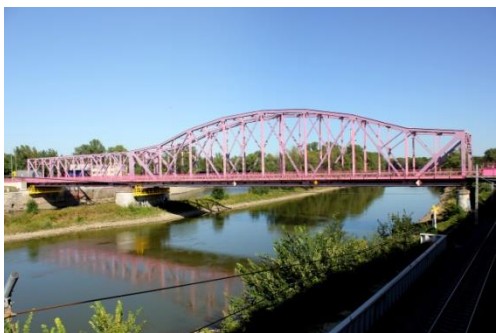

**Figure 25.** Bridge of Tolerance in Głogów, Poland (photo by Andrzej Otrębski, 2018, CC BY-SA 4.0 via Wikimedia Commons).

The pink color can also be used to symbolize peace and unity between different religions and nations. That is the case of the Masjid Dimaukom (Pink Mosque), built in 2014 in Datu Saudi Ampatuan, Maguindanao, Philippines. It is one of only three entirely pink-colored mosques in the world (the other two are the Putrajaya Mosque (Putra Mosque) in Putrajaya, Malaysia (1999), and the Great Mosque An-Nur (Masjid Besar An-Nur) in Kawal, Indonesia (2022)). Construction of the Dimaukom mosque was financed by Samsudin Dimaukom, then mayor of Datu Saudi Ampatuan, and built on property owned by his family. Christian workers built it to underline unity and interfaith brotherhood—features additionally emphasized by the pink color of the exterior. The color was chosen to represent peace and love, which Dimaukom and his wife hoped would transform the image of the Maguindanao province, sometimes marred by violence (Tuyay 2014).

Another example of the use of pink in the context of peace, love, and solidarity is an installation, Our Pink House, completed in 2016 by Polish street artist and sculptor Agata Oleksiak (also known as Olek or Crotchet Olek) in two locations—Kerava, Finland, and Avesta, Sweden. The artist used her emblematic medium—crochet—to envelop the whole building's volumes in vivid pink fabric. The art installation, realized with the help of Syrian and Ukrainian women refugees, was meant to symbolize a bright future and hope for those without a physical house (Azzarello 2016) (Figure 26).

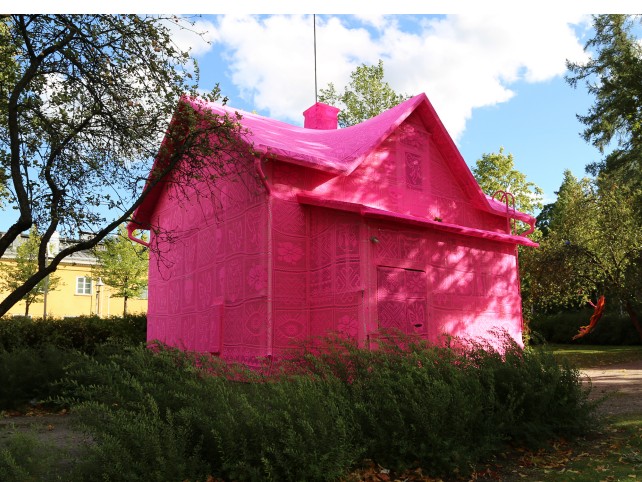

**Figure 26.** Our Pink House by Olek, Kerava, Finland (photo by Olek, 2016, image courtesy of artist).

In the next presented project, pink was used to evoke a concept of unity and play in an exceptional place. The installation, dubbed the Teeter-Totter Wall, was designed by Ronald Rael, a professor of architecture at the University of California, Berkley, and Virginia San Fratello, an associate professor of design at San Jose State University. It comprised a temporary set of fluorescent pink seesaws built along the US–Mexico steel border fence between El Paso, Texas, and Ciudad Juárez, Mexico. Although the project was in place

for only around 40 min on 28 July 2019, it was meant to foster a sense of unity at the divisive border, which was highly politicized under the then US president Donald Trump's administration. The project, inviting children on each side of the border to play together, in 2021 won the Beazley Designs of the Year award, organized by London's Design Museum every year (Walsh 2019).

*3.6. Pink as a Trendy Color—Millennial Pink*

This section will discuss the new identity of the pink color connected with the advent of the so-called "Millennial pink", which became popular towards the end of the year 2015 and, thanks to the Internet, spread widely through the world of fashion, art, and design (Bideaux 2019). It was named after "Millennials", a generation of young people born between 1990 and 2000. Additionally, in 2016, Pantone presented "Rose Quartz" as one of the two colors of the year, which was aligned with this mood and attitude (it is also worth mentioning that Pantone has proposed for color of the year also three other pinkish colors in recent years: Honeysuckle in 2011, Living Coral in 2019, and Viva Magenta in 2023).

The colors representing the term "Millennial pink" are not one specific nuance but a group of them (Bideaux 2019). These are salmon and peach shades (Y80R, Y90R, and R) rather than lavender and violet (R10B, R20B, R30B), and they are also relatively light colors with low saturation (0515, 0520) according to the Natural Color System nomenclature (Tarajko-Kowalska 2022). "Millennial pink" was a reaction against all the stereotypes associated with the pink color, especially those connected with its widely understood femininity. According to Valerie Steele, pink has increasingly been seen as a cool, emancipated, and gender-neutral color (Steele 2018). The great popularity of this phenomenon, especially among the young, initiated an evolution in the approach to pink also in architecture, where it is no longer so strongly "stigmatized" by its gendered symbolism inherited from the twentieth century. This new attitude to pink is particularly noticeable in residential buildings, whose number has significantly increased in recent years. Thanks to new technologies, it is also possible to complete entirely monochromatic pink buildings finished in mass-colored concrete, which makes the final effect even more powerful. As an example, two houses in Portugal and one in the UK can be mentioned here: Red House, designed by extrastudio in Setúbal (2016), The Pink House, by Mezzo-Atelier in Ponta Delgada, São Miguel (2017), and Seabreeze House by RX Architects in East Sussex, UK (2021).

Another modern house aesthetically belonging to the "Millennial pink" époque is the Salmen House, designed by Office S&M in 2017 and located in the Plaistow district of London, UK. The pale pink color of the façades breaks both with the traditional design and dark colors of the street context (Salmen House 2018).

In recent years, public-use buildings have also appeared in which pink is used as a modern, mature architectural color. Among them is Ensemble Mereville, a multifunctional building designed in 2018 by Depeyre Morand Architectures and situated in the center of Méréville, a small town in the French countryside. The building is finished in light pink concrete, and its color refers to the traditional pink coatings of the traditional city houses (Ensemble Mereville 2022).

Another example is The Plainfaing Tourist Office (88) by Studiolada, built in 2019 in front of the Confiserie des Hautes Vosges—a traditional confectionery and factory, which welcomes nearly 250,000 visitors a year, being one of the first "industrial" tourism locations in France. The building was designed to improve the attractiveness of the Saint-Die-des-Vosges region to tourists and encourage them to discover the area further. The cylinder-shaped volume of the building, as well as all the surrounding surfaces, were covered with pink sandstone stones that came from the Champenay quarry in Plaine (Plainfaing Tourist Office (88) (2019)).

Among the most noticeable recent pink projects is undoubtedly The Webster store in Los Angeles, CA, USA (2020), where the ground floor façade and the entrance zone were covered in pink concrete by renowned British-Ghanaian architect David Adjaye. The pink-tinted concrete was intended to both complement and contrast the grey color of the upper

part of the brutalist-style building. Adjaye, who has often experimented with materials and colors in his projects, stated that, "Pink felt like fashion, but I wanted to make something that was tough and gentle at the same time" (Gibson 2020).

*3.7. Pink as an "Instagramable" and Fictional Color*

It is impossible to understand the modern meaning of pink in architecture without its virtual aspect. There are even suggestions that the trend for "Millennial pink" itself had some connection with Wes Anderson's 2014 movie "The Grand Budapest Hotel", which was set in a fictional hotel painted on the outside in several pink colors and with mostly pink furnished interiors (Bartkowska 2021). Regardless of how accurate these suggestions are, there is no doubt that this fictional hotel is one of the most recognizable pink buildings today and that its aesthetics greatly influenced the later popularity of "Millennial pink", especially in interior design.

Just as "Millennial pink" has become popular thanks to the Internet, some pink buildings have become famous thanks to social networks such as Instagram. For example, that is the case with the Paul Smith Shop, located on Melrose Avenue in Los Angeles, CA, USA. Although it was built in 2005, five years before Instagram was created, it became well known only recently, thanks to the peculiar fashion of taking pictures in front of its bright pink façade. Made famous by the thousands of selfies and Instagram snaps taken at the #pinkwall, the giant brutalist pink building has become a local landmark as well as one of the top LA tourist attractions (Paul Smith 2023).

Nowadays, social media are so powerful that some architectural objects are even created to appear on the Internet and intended to become "Instagramable".

As an example, a temporary installation titled "Pink Is a Nice Color" can be presented. Three single-family houses scheduled for demolition were totally painted in "Pepto-Bismol" pink, together with their roofs and driveways. The artistic intervention was completed by The Mural Agency (@themostfamousartist) together with M-Rad Architecture, the company behind the demolition. The pink installation, popularized by Instagram, had thousands of visitors during its availability in 2017 (Barragan 2017).

Many graphic artists and architects also create fictional pink buildings and spaces, with some so realistically visualized that questions arise about their actual location. The most spectacular is the very recent (2023) series of digital artworks "Take Over" by Andrés Reisinger (born 1990), reimagining historical buildings in the cities of London, Rome, Paris, Tokyo, and New York with monumental, furry, pink draperies. Reisinger "added" hyper-realistic elements to existing buildings that entirely changed both their shape and the surrounding space. Pink subtly has a soothing effect, giving delicacy to everything around it (Burgos 2023).

The artist compared looking at buildings wrapped in pink installations to noticing someone who stands out in a crowd, and said: "Have you ever encountered a gentle lady in a pink fur coat? She is intriguing, a visual experience that teases a fashion show or some other type of "in real life" experience. Instead of following the white rabbit, follow the pink lady, in whatever realm she might take you" (Take Over 2023).

## 4. Conclusions

Nowadays, pink color possesses a special place in architecture, although it is still not a very popular and typical color for architectural exteriors, and its usage can only be understood in the broader socio-cultural context. There are some unique traditions associated with using pink in architecture, which are still cultivated, as in the cases of the "Pink City" of Jaipur, "Suffolk pink", or the color "Mexican pink", which has become a part of that country's cultural heritage and is used both by famous architects and in everyday architecture-without-architect. The presented selection of iconic, well-known, 20th-century buildings characterized by their distinctive use of pink color proves that pink can be used for buildings with diverse functions and forms, even skyscrapers. But at the end of the 20th century, due to stereotypical gender-coded associations and the specific

kitschy, plastic aesthetics associated with it, it became almost thoroughly discredited as a serious architectural color. Only in the last ten years has pink gradually been liberated from the stigma of the feminine, sometimes infantile or childish, "Barbie pink" loved by little girls, towards the mature "Millennial pink", an emancipated symbol of independence and universality. Since both these trends function side by side in the field of mass culture, it will probably be the same in architecture. Although modern projects show that the color pink can be used in a more ambitious and sophisticated way, idealized pink houses are still erected for fans of the "Barbie aesthetic", whose community is as strong as the supporters of the neutral "Millennial pink". This can be evidenced by the recent popularity of the fashion trend Barbiecore (2022) or the premiere of the film "Barbie" (2023). Nevertheless, thanks to the popularity of the delicate "Millennial pink", pink color has gained a new identity and has been restored to the favor of architects, who have started to use it again in their projects, albeit cautiously and for now mostly in residential buildings.

The pink color usage in contemporary architecture is very diverse and reflects the various associations and symbolism of the color itself. Some are related to the stereotypical understanding of pink and emphasize its connection with broadly understood femininity and eroticism. Others refer to childhood, meant variously as a period of innocence, infantile naivety, or girlish immaturity. Pink also appears as a contrasting color, especially in relation to the surrounding greenery, but also to the grey colors of the buildings that started to dominate modern cities in recent decades. It is also a frequent color for temporary art installations—a momentary flash of bright color like a flower blooming for just a moment. It is also often the color of extravagant residences whose eccentric owners want in that way to stand out from society. In many projects, pink is also used as a manifesto color of awareness for specific causes—mainly connected with peace, love, tolerance, and solidarity. Finally, it is a neutral and emancipated "Millennial pink" that tries to disenchant pink and restore it to maturity and sophistication.

So, probably, pink will continue to be the color that divides on the one hand and connects on the other, loved by some and hated by others—the most controversial of all the colors, but at the same time, the most pretty and powerful. The growing number of projects in pink, also in the form of digital, realistic visualizations, and popularized also thanks to social media like, e.g., Instagram, presumably shows the future of this color in architecture.

**Author Contributions:** Conceptualization, J.T.-K. and P.K.; methodology, J.T.-K.; validation, J.T.-K. and P.K.; investigation, J.T.-K. and P.K.; resources, J.T.-K. and P.K.; writing—original draft preparation, J.T.-K.; writing—review and editing, J.T.-K. and P.K.; visualization, J.T.-K.; supervision, J.T.-K. and P.K.; project administration, J.T.-K. and P.K. All authors have read and agreed to the published version of the manuscript.

**Funding:** This research received no external funding.

**Data Availability Statement:** No new data were created.

**Conflicts of Interest:** The authors declare no conflict of interest.

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
