# Peer review of "“Pretty in Pink”—The Pink Color in Architecture and the Built Environment: Symbolism, Traditions, and Contemporary Applications"

_arts, 2023_

Round 1

Reviewer 1 Report

The manuscript is interesting. The aim to study the pink colour in architecture and build environment is new and fresh. However, there is imbalance between sections 3.1. and 3.2. In section 3.1. the authors show the examples of buildings or cities, versus in 3.2. the contemporary examples are more analysed via six categories. The manuscript is quite long. The analysis of the section 3.2. (contemporary architecture) is coherent and strong. If the manuscript needs to be shortened, this article could only focus on the new architecture and environment (3.2.). 

Introduction

The balance between the sections Abstract, Introduction and Conclusions is confusing.

Materials and methods

The analysis is based on 1) literature review, 2) documentation of the colours of historical buildings, 3) illustrations and descriptions and 4) author’s own explorations in situ. Could the author write more about the analysis different types of colour data, documented (and measured?) colours and web-based images? Image based pink colour give information about the images, not the real façade colours. 

Results and discussion

The pink as an English colour name is lightly problematic. In chapter “3.1. Pink as a traditional and historical color”, the older architecture has studied via contemporary colour name “pink”. In question of the older buildings, the perceived colour is related to materials, pigments, plastering and so on. That’s why the authors could use the pinkish, reddish colour names such as the light red, rosa, flesh red etc. Also, pinkish colours are culture and language related. Then the adding of the original local colour names could help the reader understand the broader meaning. For example, in Japan, there is a word, sakura-iro, what means “the colour of cherries’ blossom”. This colour is lighter than Barbie pink. One new figure could open more the colour concepts of the older pinkish colours and contemporary pink colours. The summary of 3.1. could be showed as a table, with the façade materials, original colour names (in English and original language), and English description of pinkish colour. In this version of manuscript, the rich variation of pinkish colours (older architecture) is simplified contemporary "pink".

The writers continue Tarajko-Kowalska’s colour project (red, green, blue, yellow, white and black). It would be interesting to compare the results of those colours and pink. Then, the references of other projects or papers needed. 

Conclusions

The balance between the chapters Abstract, Introduction and Conclusions is confusing. The conclusions of Section 3.1. are missing. Also, the comparison between sections 3.1. and 3.2. could be interesting.

References

References The manuscript includes a long list of references. I did not check all of them, which references are literature references, and which reference are websites of the 100 images (data references). Are all the references used on the manuscript? All the references and the links should check once again. The website was missing in some references.

Other minor comments, 

-Does all specific information about buildings needed?

-What is the influence of Pantone’s colours Living coral (the colour of the year 2019), Rose quartz (the colour of the year 2016), Honeysuckle (the colour of the year 2011) or flamingo pink for the contemporary pink architecture?

Minor editing of English language required.

Author Response

Dear Reviewer 1,

Thank you for giving us the opportunity to submit a revised version of the manuscript “'Pretty in pink' - the pink color in architecture and the built environment: symbolic, traditions and contemporary applications" for publication in MDPI Arts.

We express our gratitude for your time and effort in improving our article. We are honored that you found our manuscript interesting. We hope that you will also appreciate the revised version.

Thank You very much for the detailed review, valuable comments, and helpful suggestions. We checked them thoroughly and attempted to incorporate them into the improved version of the article. The most important corrections made on the file are marked in blue.

In the attached file, we refer in detail to Your comments and concerns and describe how they were taken into account in the manuscript.

We hope that our paper will now meet the required publication standards.

Sincerely yours,

Authors

Reviewer 2 Report

What is the most important facts concerning the use of the pink color in the built environment, considering its symbolic, functional, and decorative aspects, with particular emphasis on Western cultures? It is interesting with relevance for contemporary architectural trends. It is up to date with current trends in architecture and popular culture. I suggest that the part dealing with architectural history and tradition should be left out. The examples are few and the conclusion is wrong. There is more to learn for the writer. The parts considering traditional architecture.I  suggest the part on pink architecture in history should be excluded. It is an interesting summary on a trend about what is going on in contemporary architecture. Thank you!   

Ok

Author Response

Dear Reviewer 2,

Thank you for giving us the opportunity to submit a revised version of the manuscript “'Pretty in pink' - the pink color in architecture and the built environment: symbolic, traditions and contemporary applications" for publication in MDPI Arts.

We express our gratitude for your time and effort in improving our article.

We are honored that you found our manuscript interesting, at least in part concerning contemporary architecture. We hope that you will appreciate the revised version.

Thank You very much for the review, valuable comments, and helpful suggestions. We checked them thoroughly and attempted to incorporate them into the improved version of the article. The most important corrections made on the file are marked in blue.

As recommended, we re-edited the article by changing the previous division into two sections of 3.1 and 3.2. The historical section (3.1) has been completely revised, and instead of being a separate subsection, it has become one of the categories for analysis. In the analysis, we focused in particular on the architecture of the 20th and 21st centuries, thanks to which the article was also shortened. We decided to keep part of the previous historical section but take into account only those traditions that are still alive and cultivated today and therefore have an impact on the current use of pink in architecture.

We hope that our paper will now meet the required publication standards.

Sincerely yours,

Authors

Round 2

Reviewer 1 Report

I would like to congratulate the authors on the changes made to the manuscript. The structure of paper and the results are more coherent. The article is ready for publishing.

Author Response

Dear Reviewer 1,

Thank you for giving us the opportunity to submit a revised version of the manuscript “’Pretty in pink’ - the pink color in architecture and the built environment: symbolic, traditions and contemporary applications” for publication in MDPI Arts.

Once again, thank You for Your reviews and supporting comments, which helped us to improve the article. We are pleased that You are satisfied with the corrections we made in the revised version of the manuscript.

Since we had a few more comments from the second reviewer, we marked them in blue in the article.

We hope that our paper will now meet the required publication standards.

Sincerely yours,

Author

Author Response

Dear Reviewer 2,

Thank you for giving us the opportunity to submit a revised version of the manuscript “’Pretty in pink’ - the pink color in architecture and the built environment: symbolic, traditions and contemporary applications” for publication in MDPI Arts.

Once again, thank You for Your reviews and supporting comments, which helped us to improve the article. We are pleased that You are satisfied with the previous corrections we made in the revised version of the manuscript. We checked Your last comments and suggestions thoroughly and attempted to incorporate them into the revised paper. The corrections made on the file are marked in blue.

In the attached file, we refer in detail to Your comments and concerns (please see the attachment.)

We hope that our paper will now meet the required publication standards.

Sincerely yours,

Authors
